# Confirmatory Study on Costs and Time Loss from Pre-Anesthetic Consultations for Canceled Surgeries: A Retrospective Analysis at Hannover Medical School, Germany

**DOI:** 10.3390/jcm14186454

**Published:** 2025-09-13

**Authors:** Steffen B. Wiegand, Anna S. Heinemann, Dennis Harries, David Bürger, Lisa Thiehoff, Anna Fischbach

**Affiliations:** 1Department of Anesthesiology and Intensive Care Medicine, Hannover Medical School, Carl-Neuberg-Straße 1, 30625 Hannover, Germany; 2Department of Anesthesiology, RWTH Aachen University, Pauwelsstraße 30, 52074 Aachen, Germany

**Keywords:** pre-anesthetic consultation, pre-anesthesia clinic, PAC, preoperative care, surgery cancellation

## Abstract

**Background/Objectives**: Pre-anesthetic consultations (PACs), conducted by anesthesiologists, are an established procedure to assess individual perioperative risk in surgical patients. Cancellation of surgery leads to unnecessary PACs, which have to be avoided in times of staff shortage. The aim of the study is to evaluate the costs and time loss of unnecessary PACs at Hannover Medical School, a university hospital in Germany. **Methods**: All PACs conducted at Hannover Medical School in September 2023 were included in the analysis. The duration and associated costs were calculated based on electronic documentation. Results were compared to data from the university hospital Aachen in Germany. **Results:** In 4.2% of all PACs, no surgical or interventional procedure was subsequently performed and therefore no anesthesiologic services were required. Reasons for surgery or intervention cancellation included choosing a conservative approach over surgery or proceeding with surgery under local anesthesia without the presence of an anesthesiologist. The additional costs for unnecessary PACs were EUR 1612 which corresponded to 43.2 h of working time. Projected over a year, this would result in EUR 19,344 in costs and 518.4 h of time spent. In comparison to university hospital Aachen, we observed lower cancellation rates after PACs and a greater reliability of the planned interventions and surgeries. **Conclusions**: The results of this study suggest that the financial and time burden associated with these consultations was not substantial. To ensure optimal use of temporal and financial anesthesiologic resources, it is essential to avoid PACs for patients who will not undergo surgery or intervention.

## 1. Introduction

The pre-anesthetic consultation (PAC) is a critical step in the preparation for surgery. Over the past 20 years, the number of hospitals in Germany has decreased by approximately 20%, while the number of surgical procedures has increased to more than 16 million annually [1,2]. Consequently, the demand for PACs is substantial, as every anesthetic procedure requires a prior consultation. In 2024, more than 28,000 anesthesiologists were working in Germany and were responsible for conducting these consultations [3].

PACs have been shown to offer numerous benefits for patients undergoing surgery. They can reduce perioperative morbidity and mortality and lower the rate of surgical cancellations due to insufficient preoperative preparation [4,5,6,7]. Moreover, PACs have been associated with shorter hospital stays [8]. The costs for PACs account for approximately 4% to 9% of the total anesthesia costs [9,10]. Overall, PACs are widely recognized as an essential component of modern anesthesia care. However, surgery cancellations that occur independently of the risk assessment during PACs lead to additional costs and time losses. Previous studies have reported cancellation rates ranging from 2.3% to 11.8% [8,9,11]. More recently, Simons et al. found a cancellation rate of 7.8% at a university hospital in Germany but were unable to determine the reasons for nearly half of these cancellations [12]. Staff shortages are expected to worsen in the coming years and will create significant challenges for optimizing preoperative processes [13]. Avoiding unnecessary PACs is therefore crucial to ensure efficient staffing management.

The aim of this study is to evaluate the costs and time losses associated with PACs in a large university hospital in Germany and to analyze cases in which patients underwent PACs but ultimately did not proceed to surgery or intervention.

## 2. Materials and Methods

### 2.1. Ethical Approval

This retrospective study was conducted in accordance with the guidelines of the Declaration of Helsinki, the principles of Good Clinical Practice, and the standards of the local ethics committee. Data were fully anonymized prior to analysis. The local ethical committee of Hannover Medical School approved this retrospective analysis of patient data (11296_BO_K_2024, 03/24).

### 2.2. Study Groups

This study adopted the methodology described by Simons et al. [12]. The analysis was based on data collected from the PAC clinic at Hannover Medical School during September 2023. All PACs were documented using ANDOK live (Version 2.9.4.2; DATAPEC Medical Solutions, Pliezhausen, Germany). Prior to analysis, all patient data were pseudonymized. Patients were categorized into two groups: ‘ANE,’ comprising those who received anesthetic services (e.g., anesthesia for surgical procedures, sedation for MRI scans), and ‘NoANE,’ consisting of patients whose PAC was not followed by any anesthetic intervention.

The term “elective” refers to planned interventions that are scheduled in advance because they do not involve a medical emergency. In contrast, “non-elective” refers to urgent interventions that must be performed within 24 h.

“Conservative therapy” describes non-interventional and non-surgical approaches, such as pharmacological treatment or watchful waiting, often chosen due to high procedural risks.

“Double consultation” refers to performing a second PAC, either because the initial one has expired or because an unnecessary additional consultation was conducted by mistake.

The term “bedside ICU procedure” refers to non-surgical interventions that were performed in the intensive care unit to prevent the risks involved in transferring critically ill patients to the operating room.

Second anesthesiologist refers to the consultation of another anesthesiologist during PAC. This was the case when the attending anesthesiologist was uncertain about how to proceed. Most of the time, the second anesthesiologist was of higher training level and consultation was performed via telephone support. In some cases, support was given in person in form of supervision of the first anesthesiologist for training reasons.

### 2.3. Inclusion and Exclusion Criteria

Patients who visited the PAC clinic, either as outpatients or as inpatients already admitted to the hospital, were included in this study. PACs were conducted by anesthesiologists either in the PAC clinic or, for patients with mobility limitations, on the wards. We excluded all cases registered by surgical colleagues as immediate operations or as very urgent procedures requiring intervention within less than two hours, because such time-sensitive situations do not align with the patient-oriented approach of a PAC.

### 2.4. Cost and Time Calculations

Cost calculations were performed according to the methodology described by Simons et al. [12]. The calculation for PAC costs was primarily based on the remuneration table applicable to physicians at German university hospitals in September 2023 (see Appendix A). Costs related to rent, administrative staff salaries, office supplies, or infrastructure were excluded, as these are difficult to allocate to individual cases. The time required for a PAC was defined as the period from the initial face-to-face contact with the patient—or the start of chart review—until the completion of documentation.

### 2.5. Statistical Analysis

Statistical tests were performed using GraphPad Prism (v.8.4.3.; San Diego, CA, USA). Data are expressed as mean ± SD. Normal distribution was tested with the Shapiro–Wilk test. As none of the data sets were normally distributed, only non-parametric tests were used for analysis. The Mann–Whitney–U test with two-tailed *p*-value was used to compare two groups, and the Kruskal–Wallis test with Dunn’s correction was used for multiple comparisons of more than two groups. The chi-squared test was calculated for the comparison of discrete variables if all cells had expected frequencies of ≥10. If frequencies were ≤10, Fisher’s exact test was used. To analyze predictors for cancellation, a binary logistic regression analysis was performed by using SPSS software (Version 22, SPSS Inc., Chicago, IL, USA). The inclusion of parameters for the logistic regression model was based on results from the Mann–Whitney test, chi-squared test, or Fisher’s exact test. Parameters with *p*-values of <0.1 were included. A Kaplan–Meier curve was plotted for better visualization of the outstanding procedures after the PACs. Other measures of central tendency and variability or reliability are available in the supplements (additional descriptive statistics.pdf). A *p*-value < 0.05 was considered significant (* *p* < 0.05; ** *p* < 0.01; *** *p* < 0.001).

## 3. Results

### 3.1. Patient Inclusion and Exclusion Criteria

All patients who were scheduled for surgery or intervention that led to a request for a PAC were included (*n* = 2447) (Figure 1). Nine cases were excluded, as digital ANDOK live PAC protocols were created, but not completed. Patients were also excluded if they required immediate surgery (*n* = 120), urgent surgery within two hours (*n* = 165), or if the duration of the consultation was not recorded (*n* = 88). Consequently, 2065 cases were included in the final analysis. Two groups were defined: patients who received anesthetic services (ANE; *n* = 1979) and those who did not (NoANE; *n* = 86).

### 3.2. Patient Demographics and Clinical Characteristics

An overview of patient demographics and characteristics, categorized into the ANE and NoANE groups, is shown in Table 1. The median ASA score was higher in patients in the NoANE group, although the difference was not statistically significant compared to the ANE group. However, an ASA IV score was observed significantly more often in the NoANE group than in the ANE group (15.1% vs. 6.9%; *p* = 0.004). The interval between the PAC and the initially scheduled date for surgery or intervention was significantly longer in the NoANE group compared to the ANE group (median [IQR]: 3 [1; 10] days vs. 1 [1; 3] days; *p* < 0.001).

It should be noted that only a portion of the study population could be included in this analysis, as no intervention date was defined during the PAC for 228 patients in the ANE group and for 23 patients in the NoANE group. The proportion of PACs that were follow-up visits for repetitive surgery or intervention after the initial procedure was already performed was higher in the NoANE group than in the ANE group (10.5% vs. 7.9%; *p* = 0.050).

A second anesthesiologist was involved in the PAC significantly more frequently in the NoANE group compared to the ANE group (15.1% vs. 4.4%; *p* < 0.001). Furthermore, a medium cardiac risk score [14] was observed more frequently in the NoANE group than in the ANE group (55.8% vs. 44.1%; *p* = 0.033). Other characteristics, such as age, sex assigned at birth, and the proportion of elective procedures or procedures scheduled within 24 h, did not differ significantly between the groups.

### 3.3. Training Levels of Anesthesiologists and Duration of PACs

Most PACs were performed by anesthesia specialists (*n* = 448; 21.7%), followed by third-year residents (*n* = 387; 18.7%), fourth-year residents (*n* = 367; 17.8%), attending anesthesiologists (*n* = 365; 17.7%), second-year residents (*n* = 213; 10.3%), first-year residents (*n* = 158; 7.7%), and fifth-year residents (*n* = 127; 6.2%) (Figure 2A). The overall duration of PAC varied significantly depending on the ASA scores of patients (*p* < 0.001) (Figure 2B).

With increasing ASA scores, the mean duration of PACs increased. PACs for ASA IV patients had a mean duration of 29.8 ± 15.4 min, which was significantly longer than for ASA I patients (19.9 ± 9.6 min; *p* < 0.001) or ASA II patients (24.8 ± 12.0 min; *p* < 0.001), but did not differ significantly from PACs for ASA III patients (27.8 ± 14.7 min; *p* = 0.88). The mean duration of PACs also varied significantly depending on the training level of the anesthesiologist performing the consultation (*p* < 0.001) (Figure 2C).

PACs performed by first-year residents had the longest mean duration (32.1 ± 17.0 min) and were significantly longer than those performed by second-year residents (26.5 ± 13.6 min; *p* = 0.02), third-year residents (23.4 ± 12.7 min; *p* < 0.001), fifth-year residents (25.3 ± 11.8 min; *p* = 0.050), specialists (22.9 ± 12.1 min; *p* < 0.001), and attending anesthesiologists (26.5 ± 12.6 min; *p* = 0.04). However, they were comparable in duration to PACs performed by fourth-year residents (27.9 ± 14.1 min; *p* = 0.64).

Overall, the mean duration of PACs was 25.7 ± 13.5 min, resulting in costs of EUR 16.45 ± 8.97 per case. The duration of PACs varied depending on the referring medical discipline (Table 2). The longest PAC durations were observed in patients scheduled for thoracic surgery (30.6 ± 13.5 min), while the shortest were recorded for patients from psychiatry (13.8 ± 14.4 min). Consistent with these findings, PAC costs were higher in thoracic surgery (EUR 20.00 ± 7.89) and lower in psychiatry (EUR 9.13 ± 7.78).

### 3.4. Overall Duration and Costs of PACs

Of all PACs performed (*n* = 2065), most led to the requested anesthesiology service (*n* = 1979; 95.8%). Only a small number were cancelled (*n* = 86; 4.2%) (Figure 3A). Cancellations occurred for various reasons. In many cases, the procedure was cancelled because a conservative treatment approach was chosen (*n* = 24; 27.9%), either due to high surgical or procedural risk or if intervention was no longer indicated.

In some cases, patients required a second PAC (*n* = 16; 18.6%) because the initial assessment had expired. In other cases, surgery was performed under local anesthesia by the surgeon (*n* = 16; 18.6%), or obstetric patients delivered spontaneously without epidural anesthesia (*n* = 12; 14.0%). Less frequent reasons for cancellation included the patient’s own decision (*n* = 4; 4.7%), capacity constraints (*n* = 4; 4.7%), or delays due to acute illness (*n* = 2; 2.3%). In a few cases, the reason for cancellation remained unknown (*n* = 5; 5.8%).

Total costs for all PACs performed in September 2023 (*n* = 2065) were EUR 31,958, requiring 885.8 h of time (Figure 3B,C). In 4.2% of PACs, no anesthesiological service was subsequently needed. These NoANE cases accounted for EUR 1612 (5.0% of all PAC costs) and 43.2 h (4.9% of all PAC time). Projected over a year, this would result in EUR 19,344 in costs and 518.4 h of time spent.

### 3.5. Missing Test Results

Neither group required a second PAC visit due to missing test results from the initial consultation. Overall, missing preoperative test results were significantly less frequent in the ANE group compared to the NoANE group (210 [10.6%] vs. 16 [18.6%]; *p* = 0.020) (Table 3). The rate of missing preoperative ECGs was also lower in the ANE group than in the NoANE group (81 [4.1%] vs. 8 [9.3%]; *p* = 0.029). There were no statistically significant group differences regarding the availability of preoperative diagnostic test results, including laboratory tests, echocardiographic evaluations, imaging procedures, or consultations from cardiology, pulmonology, or other specialties.

To identify predictive variables for procedure cancellation, a binary logistic regression model which included three parameters was used. The model was highly statistically significant (χ^2^(3) = 18.425; *p* < 0.001) and correctly classified 95.8% of cases (Table 4). A strong predictor of cancellation was the involvement of a second anesthesiologist during PAC, which was associated with an almost fourfold increase in cancellation risk (OR: 3.823; 95% CI: 2.035–7.182; *p* = 0.003). By contrast, medium cardiac risk score or if the PAC was already a second PAC was not significantly associated with an increased risk of cancellation in the analysis (OR: 1.585; 95% CI: 1.022–2.456; *p* = 0.078 and OR: 1.316; 95% CI: 0.644–2.689; *p* = 0.451).

In 77.7% of all cases, the scheduled date for surgery or intervention matched the actual date on which the procedure was performed. Discrepancies are illustrated in the Kaplan–Meier curve shown in Figure 4. A difference of more than three days occurred in over 10% of cases.

## 4. Discussion

In September 2023, 2065 pre-anesthetic consultations (PACs) were conducted, with 86 (4.2%) not leading to procedures requiring anesthesia care providers (NoANE group). Compared to other patients, those in the NoANE group more often had an ASA IV status and experienced longer intervals between the PAC and the initially scheduled procedure. ASA IV status is associated with increased morbidity, and therefore, a conservative treatment approach seems to be preferred due to increased anesthesiological risk. Additionally, these consultations were more frequently follow-up consultations after prior surgeries or interventions, and they more often involved a second anesthesiologist and were more commonly associated with procedures of intermediate cardiac risk.

Further, this study identified several reasons why PACs did not lead to subsequent procedures, including the choice to perform surgery under local anesthesia, repeated PAC visits for the same patient, and shifts toward conservative treatment strategies. Nearly 5% of the total costs and time associated with PACs were related to the NoANE group. The risk of cancellation was almost four times higher when a second anesthesiologist was involved in the PAC and increased by 58% if the planned procedure was classified as intermediate-risk surgery. In approximately 80% of cases, the scheduled date of surgery or intervention matched the actual procedure date. Moreover, about 90% of procedures were performed within three days of the originally scheduled date.

Projected over an entire year, the additional costs are estimated at EUR 19,344, with an associated time loss of 518.4 h. Although these cost projections may seem relatively low and manageable in the context of a large hospital, they should be interpreted cautiously. The present calculation considers only physician salaries and excludes costs for infrastructure, administrative personnel, utilities, office supplies, and other overheads. Thus, the actual costs related to PACs are likely to be considerably higher. Data from previous studies illustrate the range of such costs: as early as 1995, a cost analysis reported expenses of £20.60 per PAC [15]. A study in Hong Kong calculated costs at $109 per case [9], while costs in a German university hospital were reported between EUR 29 and EUR 38 per PAC [12]. At our institution, PAC services are not billed separately but are instead incorporated into an overall anesthesia billing.

Compared to the findings of Simons et al., our study found a slightly shorter mean duration of PACs, averaging 1.4 min less per case and leading to a modest cost reduction of EUR 0.49 per PAC. Despite the small magnitude of this difference, it is notable, as our cohort included nearly 10% fewer ASA I patients.

This variation may reflect the higher proportion of PACs performed by specialists and attending anesthesiologists in our institution relative to the cohort studied by Simons et al. Interestingly, whereas Simons et al. reported that first-year residents required the least time to perform PACs—a finding that appears counterintuitive—our study found that first-year residents needed more time.

It is plausible that more experienced anesthesiologists, with their deeper knowledge of anesthesia-related pathophysiology and greater familiarity with the PAC process, can perform these consultations more efficiently. Another potential explanation lies in the differences between the documentation systems. In the study by Simons et al., anesthesiologists use a two-step documentation process, initially recording patient information on paper and subsequently transferring it into a digital format. By contrast, our study primarily employed digital documentation. The implementation of a fully electronic system may enhance the efficiency of the PAC process.

The future of inpatient and outpatient PACs will probably combine video-based patient education with personalized consultations. After watching an educational video, patients can discuss individual risks and address any specific questions or concerns with the anesthesiologist. During the COVID-19 pandemic, virtual care and video consultations gained significant popularity [16,17]. Currently, video-based patient education for PAC is employed in rural areas and has demonstrated patient satisfaction rates comparable to those of traditional in-person education [18,19]. Video-based PACs may enhance process efficiency, as general information regarding anesthesia can be effectively provided through video content. This approach has the potential to reduce the time physicians spend on basic explanations, enabling them to focus more closely on assessing individual patient risk factors.

Additionally, video-based outpatient PACs could reduce CO_2_ emissions by eliminating unnecessary hospital visits [20], which might also be achieved when PAC is scheduled as part of a necessary surgery visit.

In this study, no scheduled second PAC visits were required, representing a significant difference from the findings reported by Simons et al. [12]. In cases where the anesthesiologist performing the PAC was uncertain about how to proceed, a second anesthesiologist was consulted to support the risk assessment. If additional diagnostic tests were requested, no second consultation was scheduled; instead, the senior anesthesiologist responsible for the respective surgical department was informed about the pending results and reviewed them prior to the planned intervention or surgery. This approach did not result in substantially higher costs, as demonstrated in our analysis, yet it contributed to optimizing the PAC process and effectively minimized the need for additional PAC visits.

While Simons et al. reported overall missing preoperative results in about 30% of all requested PACs, we observed only 10% missing preoperative results. In general, the adherence to pre-operative anesthesiologists’ guidelines is quite low, as it has been demonstrated before [21]. In this study, we could not differentiate if the difference in missing result rate was real or caused by underreported missing preoperative results or shifts from current anesthesiologic recommendations for pre-operative assessment.

Studies have demonstrated that the implementation of PACs has contributed to a reduction in surgery cancellation rates [22,23]. To further decrease cancellations and avoid unnecessary PACs, it is crucial to analyze the underlying reasons for such cancellations. Wongtangman et al. identified a history of anxiety disorder as the most important predictor for surgery cancellation [24]. In the present study, a specific cause could be identified for 95% of all cancellations. One of the most common reasons was that surgeries were performed under local anesthesia without requiring the involvement of the anesthesiology department. Notably, 70% of these cases originated from the ophthalmology department, where numerous eye surgeries can be conducted without sedation or general anesthesia [25].

Second PACs were the other most frequent reason for unnecessary PACs in the NoANE group. The reason for those PACs was that the last consultation was no longer valid due to prolonged time between the last PAC and the planned procedure or another PAC was scheduled without medical necessity by mistake. These potentially avoidable PACs could be reduced through improved coordination, whereas other reasons for procedure cancellation—such as patient decisions or institutional capacity constraints—are likely less predictable and more difficult to influence.

Overall, our institution demonstrated greater reliability in maintaining scheduled surgical dates following PACs compared to the findings reported by Simons et al. [12]. Nearly 80% of patients underwent their procedures on the planned date, and slightly more than 10% of cases remained pending after three days. In contrast, in the study by Simons et al., it took 24 days for 90% of procedures to be completed.

This study has several limitations. First, due to its retrospective design, there is a risk of incomplete or inaccurate documentation. Second, this is a single-centre study, which may limit the generalizability of the results to other institutions. Third, we limited our analysis to the same month studied by Simons et al. [12] to enhance comparability. However, focusing on one month may introduce bias if that month is an outlier, whereas analyzing a longer period could help to minimize this risk.

Nevertheless, this study highlights an often-neglected topic in clinical practice. Although our results indicate that the financial and time losses associated with unnecessary PACs were not substantial, it remains important to minimize uncertainty about whether anesthesiology service is truly needed for certain procedures. Reducing such uncertainties is essential, as it helps to avoid unnecessary consultations and allows anesthesiologists to focus on patients who genuinely require specialized anesthesia services.

## Figures and Tables

**Figure 1 jcm-14-06454-f001:**
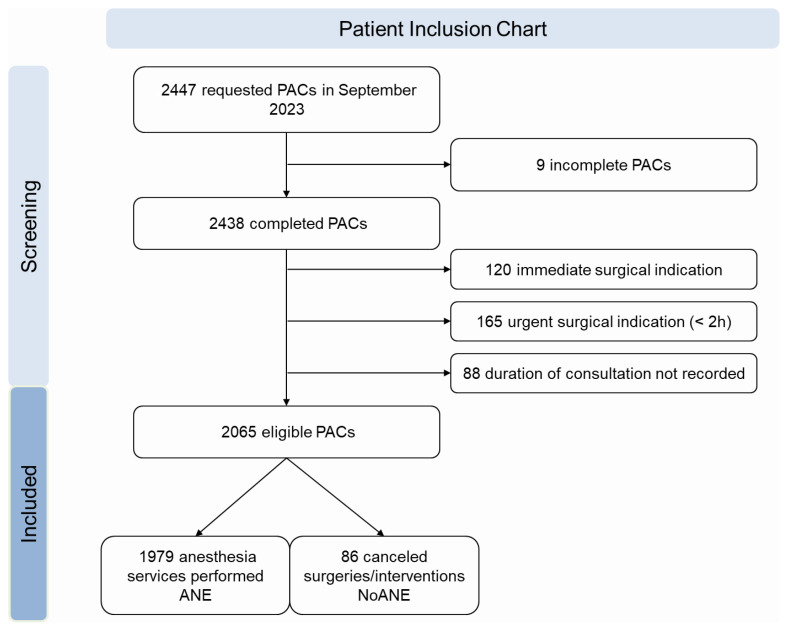
Patient screening and inclusion flowchart.

**Figure 2 jcm-14-06454-f002:**
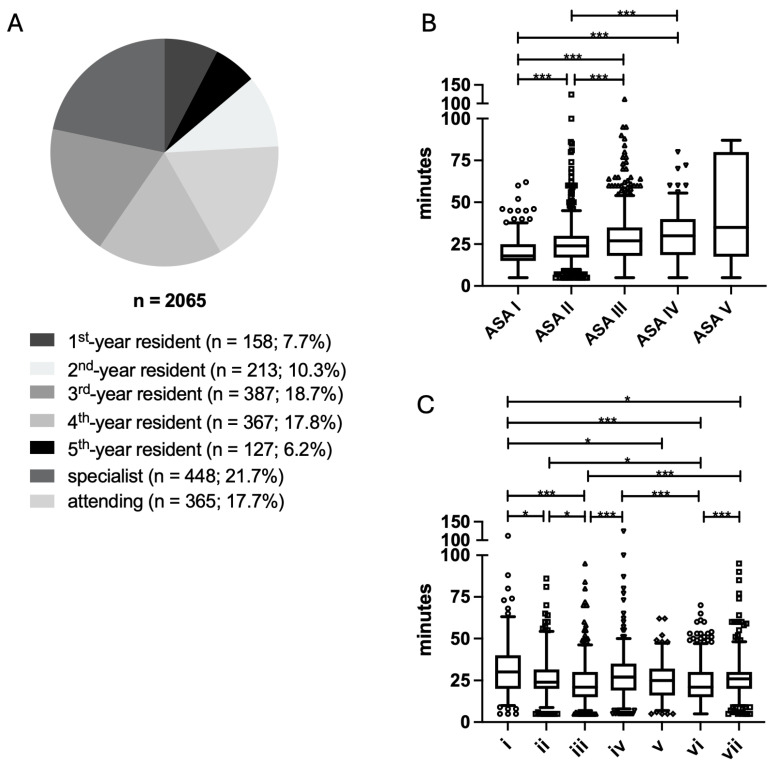
(**A**) Number of PACs performed by anesthesiologists, categorized by residency year. (**B**) Duration of PACs in relation to the ASA physical status classification of patients. (**C**) Duration of PACs in relation to the level of training of the anesthesiologist. Results are expressed as median (95% CI). i = 1st-year resident; ii = 2nd-year resident; iii = 3rd-year resident; iv = 4th-year resident; v = 5th-year resident; vi = specialist; vii = attending physician; * *p* < 0.05; *** *p* < 0.001.

**Figure 3 jcm-14-06454-f003:**
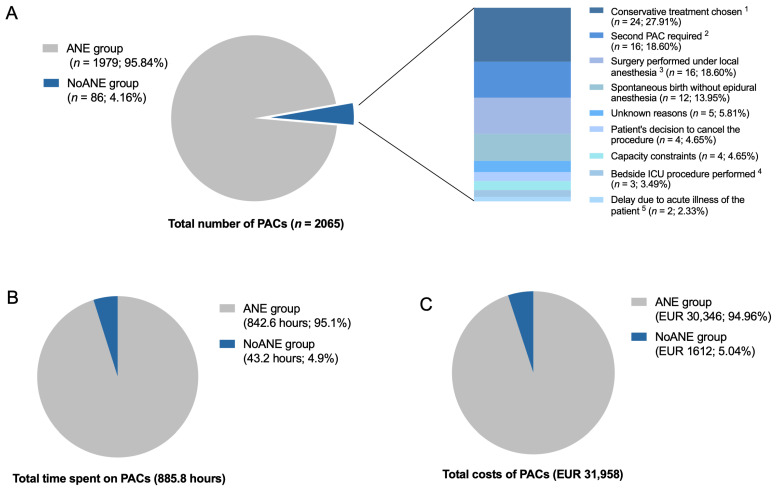
(**A**) Number of PACs in the ANE and NoANE groups, including reasons for cancellation in the NoANE group. (**B**) Duration of PACs in the ANE and NoANE groups. Results are expressed as number (%) of cases. (**C**) Total costs of PACs in the ANE and NoANE groups. ^1^ A conservative treatment approach was chosen because of high anesthesiological risk or if intervention was no longer indicated. ^2^ An additional PAC was performed either because the validity period of the initial assessment had expired or because it was mistakenly scheduled. ^3^ Surgery was performed under local anesthesia by the surgeon without the need for anesthesia service. ^4^ Bedside ICU procedure was performed without the need for anesthesia service. ^5^ Anesthetic service was postponed due to acute illness of the patient.

**Figure 4 jcm-14-06454-f004:**
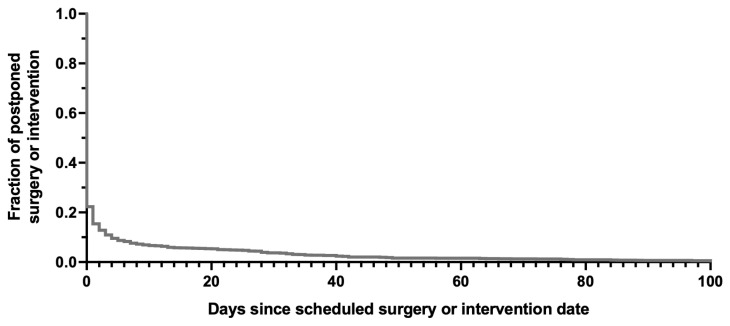
Discrepancy between the scheduled and actual dates of anesthesia services. In approximately 78% of cases, the planned date for anesthesiological care matched the actual date of service. Nearly 90% of all procedures were performed within three days of the originally scheduled date.

**Table 1 jcm-14-06454-t001:** Demographics and clinical characteristics of the study population.

Characteristic	All Patients *n* = 2065	ANE Group *n* = 1979	NoANE Group *n* = 86	*p*-Value
Age, years	50 (31;70)	50 (31;69)	53 (35;76)	0.359
Sex assigned at birth (female/male)	1001/1064 (48.5%/51.5%)	952/1027 (48.1%/51.9%)	49/37 (57.0%/42.0%)	0.107
ASA Score	2 (2;3)	2 (2;3)	3 (2;3)	0.174
ASA I	246 (11.9%)	236 (11.9)	10 (11.6%)	0.934
ASA II	893 (43.2%)	860 (43.5%)	33 (38.4%)	0.352
ASA III	772 (37.4%)	742 (37.5%)	30 (34.9%)	0.624
ASA IV	149 (7.2%)	136 (6.9%)	13 (15.1%)	0.004
ASA V	5 (0.2%)	5 (0.3%)	0 (0.0%)	>0.999
Interval between PAC and initially scheduled surgery date, days	1 (1;3)	1 (1;3)	3 (1;10)	<0.001
Current PAC is a follow-up visit	185 (9.0%)	176 (8.9%)	9 (10.5%)	0.050
2nd anesthesiologist involved in PAC	100 (4.8%)	87 (4.4%)	13 (15.1%)	<0.001
Cardiac risk score [14]	low (low; medium)	low (low; medium)	medium (low; medium)	0.303
Low	1048 (50.8%)	1011 (51.1%)	37 (43.0%)	0.143
Medium	921 (44.6%)	873 (44.1%)	48 (55.8%)	0.033
High	96 (4.6%)	95 (4.8%)	1 (1.2%)	0.183
Elective vs. non-elective surgery or intervention (24 h)	1707/360 (82.7%/17.3%)	1698/281 (85.8%/14.2%)	9/79 (10.5%/89.5%)	0.150

Results are expressed as median (25th Quartile; 75th Quartile) or No. (%). PAC = pre-anesthetic consultation, ASA = American Society of Anesthesiology.

**Table 2 jcm-14-06454-t002:** Duration and costs of PACs depending on medical specialty.

Specialty	All Patients *n* = 2065	Duration of PAC [min/Case]	Cost of PAC [EUR/Case] ± SD
Thoracic Surgery	36 (1.7%)	30.6 ± 13.5	20.00 ± 7.89
Dentistry/Maxillofacial Surgery	114 (5.5%)	29.4 ± 14.5	19.20 ± 10.80
Vascular Surgery	17 (0.8%)	29.2 ± 16.9	19.86 ± 12.92
Gynecology	269 (13.0%)	26.6 ± 10.9	19.79 ± 9.13
Internal Medicine	74 (3.6%)	30.4 ± 14.3	18.98 ± 8.55
Cardiac Surgery	168 (8.1%)	28.2 ± 12.7	18.91 ± 8.14
Neurosurgery	138 (6.7%)	30.0 ± 16.6	17.19 ± 9.50
Urology	134 (6.5%)	26.3 ± 11.4	16.35 ± 7.97
Trauma Surgery	223 (10.8%)	26.9 ± 16.7	15.68 ± 10.02
General Surgery	182 (8.8%)	25.4 ± 15.1	15.66 ± 9.46
Neurology	4 (0.1%)	25.8 ± 5.9	15.25 ± 3.34
Plastic Surgery	99 (4.8%)	24.5 ± 13.0	14.97 ± 9.40
Otorhinolaryngology	166 (8.0%)	24.0 ± 10.5	14.16 ± 6.35
Dermatology	10 (0.5%)	24.2 ± 12.1	13.94 ± 6.07
Ophthalmology	264 (12.8%)	22.5 ± 11.4	13.81 ± 7.53
Pediatrics	130 (6.3%)	19.1 ± 8.6	13.49 ± 6.39
Nuclear Medicine/Radiology	16 (0.8%)	19.3 ± 12.5	12.51 ± 8.08
Psychiatry	21 (1.0%)	13.8 ± 14.4	9.13 ± 7.78
Overall	2065 (100%)	25.74 ± 13.5	16.45 ± 8.97
*p*-Value		<0.001	<0.001

Results are expressed as mean ± SD or No. (%). PAC = pre-anesthetic consultation.

**Table 3 jcm-14-06454-t003:** Pending preoperative diagnostic test results.

Type of Outstanding Preoperative Test Results	All Patients *n* = 2065	ANE *n* = 1979	No ANE *n* = 86	*p*-Value
Total	226 (10.9%)	210 (10.6%)	16 (18.6%)	0.020
Lab test results	134 (6.5%)	126 (6.3%)	8 (9.3%)	0.263
ECG	89 (4.3%)	81 (4.1%)	8 (9.3%)	0.029
TTE/TEE	11 (0.5%)	10 (0.5%)	1 (1.2%)	0.374
X-ray/CT scan	2 (0.1%)	2 (0.1%)	0 (0%)	>0.999
Cardiology consultation	9 (0.4%)	7 (0.3%)	2 (2.3%)	0.051
Pulmonary function tests	4 (0.2%)	4 (0.2%)	0 (0%)	>0.999
Other specialty consultations	9 (0.4%)	9 (0.5%)	0 (0%)	>0.999

Results are expressed No. (%). ECG = electrocardiogram; TTE = transthoracic echocardiography; TEE = transesophageal echocardiography; CT = computed tomography.

**Table 4 jcm-14-06454-t004:** Predictors of procedure cancellation identified by binary logistic regression.

Characteristic	OR (95% CI)	*p*-Value	Adjusted *p*-Value Bonferoni Holm
2nd anesthesiologist involved in the PAC	3.823 (2.035 to 7.182)	<0.001	0.003
Medium cardiac risk score	1.585 (1.022 to 2.456)	0.039	0.078
2nd PAC required ^1^	1.316 (0.644 to 2.689)	0.451	0.451

PAC = pre-anesthetic consultation; ^1^ Analyzed PAC was already a 2nd PAC.

## Data Availability

The datasets generated and/or analyzed during the current study cannot be made openly available due to sensitive or confidential information and are available from the corresponding author upon reasonable request.

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
