# Peer review of "Confirmatory Study on Costs and Time Loss from Pre-Anesthetic Consultations for Canceled Surgeries: A Retrospective Analysis at Hannover Medical School, Germany"

_jcm, 2025, doi:10.3390/jcm14186454_

Round 1

Reviewer 1 Report

Comments and Suggestions for Authors

MAJOR

Line 146 "the rate at which a second anesthesiologist". This situation is explained at lines 367-369. Data needs to be provided on what was predictive, not the individual anesthesiologists, but rather a second anesthesiologist.

Line 246: Add the count of all predictors considered for inclusion in Table 4 based on the criteria given in line 113. Use Bonferroni or Holm-Bonferroni adjustment for the correlated multiple comparisons.

I question lines 32 and "To ensure optimal use of anesthesiologic resources, it is essential to avoid PACs for patients who will  not undergo surgery or intervention." Figure 3A shows that the lowest cost solution is often no surgery.

Please add a supplemental search for other similar articles. I used Google Scholar based on the authors' wording in lines 268 and 274:
("preoperative evaluation" OR "preanesthesia evaluation") ("not leading to procedures" OR "did not lead to subsequent procedures" OR "procedure was not performed"  )

MINOR issues of writing

Line 255 "2nd anesthesiologist involved in the PAC." Provide the raw counts 

Line 106 "Normal distribution was tested with the Shapiro–Wilk test." However, I did not find where it was tested in the article.

Line 108 "Dunn's correction." However, the multiple locations where adjustment for multiple comparisons is needed seem unstated when applied (e.g., Figure 2).

Line 189 "costs were highest" but that was not tested while adjusting the multiple comparisons. Remove statements of highest and equivalent when untested. Consider limiting the focus to the tables. There is no need for so much inferential testing unrelated to the primary question. 

Line 198 "in most cases" which means >50%  but it was 27.9%. The authors probably mean "many."

Line 249 "The strongest" is another example like that of the preceding two comments.

Reviewer 2 Report

Comments and Suggestions for Authors

Thank you for permitting me to review this manuscript 

In this study the authors assessed the  number and cost of PAC  on cancelled surgeries with the supervision of anestjhesiologists  the different reasons for cancellartions are reported and they conclude that the impact is not massive with a rate of 4.7 % rate cancellation 

here are my suggestions 

the topic is interesting however some clarifications  are needed 

for ex some cancellation may be due to PAC itself especially  for ASA  4  patients , in which anesthesia can be counterindicated and alternatrively the procedure may go forward with local anesthesia

In many countries PAC are mandatory and this is a quality assurance  and security issue therefore focus should be made upon demands and anesthesia demand should only occurr when surgery under anesthesia or even with anesthesia surveillance only if no other option is available 

as cited fairly by the authors this is a single center retrospective study and the results should be regarded very cautiously  

carbon saving may not be an issue as PAC  can be set up  after a surgery visit 

Round 2

Reviewer 1 Report

Comments and Suggestions for Authors

Comment 1. Please explain why there needs to be a second anesthesiologist. This has not been explained in a manner that is generalizable to other organizations. At line 323 in the discussion, “In cases where the anesthesiologist performing the PAC was uncertain about how to proceed, a second anesthesiologist was consulted to support the risk assessment.” Move or add to the Methods. Provide more information about “support.” Is this based on medical evidence? Provide the information that would be sufficient for these statements to be reproducible.

Comment 2. First, the number of predictors used to generate Table 4 is not given. Second, the authors write “the associated confidence intervals and p-values already reflect a form of simultaneous inference at the nominal significance level … e.g.,: Mundfrom, D. J., Shaw, D. G., & Ke, T. L. (2006). Bonferroni Adjustments in Tests for Regression Coefficients. Multiple Linear Regression Viewpoints, 32(1), 1–10.).”
Thank you for including the reference. The paper does not demonstrate what the authors claim it does. Because the authors have not specified the number of predictors used to generate their Table 4, I cannot refer to a particular row from Mundfrom et al. Table 1. Tentatively, I use those with eight predictors. For a sample size of 300, the Type I error rate is not 0.05 but 0.3249! Thus, some adjustment is needed. As the authors write, however, the Bonferroni adjustment is overly conservative. With two of the eight significant (i.e., evaluating Type II error), the Bonferroni-adjusted rate is 0.0356, not 0.05. That is why I had suggested the Holm-Bonferroni method. Any stepwise method for which simulations have shown adjustment to close to the nominal rate seems appropriate. No adjustment is not suitable based on the reference that the authors provided.

Comment 3. I wrote “I question lines 32 and "To ensure optimal use of anesthesiologic resources, it is essential to avoid PACs for patients who will not undergo surgery or intervention." Figure 3A shows that the lowest cost solution is often no surgery.” The authors state “Of note, the aim of this study was not to determine which treatment approach leads to the lowest cost but to elute the current PAC situation and cancellation reasons.” Then, the authors’ conclusion is not based on the study. I am quoting the last sentence of the abstract.

Comment 4. I gave the authors the search protocol: 
 ("preoperative evaluation" OR "preanesthesia evaluation") ("not leading to procedures" OR "did not lead to subsequent procedures" OR "procedure was not performed" ). Please address the related articles. The authors explain that “The focus on the anesthesiological aspect of the evaluation is rather underrepresented.” The authors’ point is reasonable. Consider the narrower search:
("preoperative evaluation" OR "preanesthesia evaluation") ("not leading to procedures" OR "did not lead to subsequent procedures" OR "procedure was not performed" ) ("anesthesiologists" OR "anaesthesiologists" OR "anesthetists" OR "anaesthetists")
The first two articles obtained seem related, one for veterinary anaesthesia and the other for human. 

Regarding the Shapiro-Wilk test, please add a sentence or phrase explaining how the results were applied. If the authors consistently apply one test or the other based on the result, then say that. Please see that this is not stated at line 106. Consider simply deleting the sentence about the Shapiro-Wilk test.

Author Response

Point-by-point response to reviewer

We highly thank all reviewers for taking the time to revise our work again, and the valuable input to improve substantially the manuscript. According to the comments below, we have revised the manuscript and highlighted all changes of the manuscript text in yellow.

Comment 1. Please explain why there needs to be a second anesthesiologist. This has not been explained in a manner that is generalizable to other organizations. At line 323 in the discussion, “In cases where the anesthesiologist performing the PAC was uncertain about how to proceed, a second anesthesiologist was consulted to support the risk assessment.” Move or add to the Methods. Provide more information about “support.” Is this based on medical evidence? Provide the information that would be sufficient for these statements to be reproducible.

Answer: We thank the reviewer for notifying us that the topic of second anesthesiologist requires further clarification. Therefore, we added a paragraph to the Method section as suggested.

Comment 2. First, the number of predictors used to generate Table 4 is not given. Second, the authors write “the associated confidence intervals and p-values already reflect a form of simultaneous inference at the nominal significance level … e.g.,: Mundfrom, D. J., Shaw, D. G., & Ke, T. L. (2006). Bonferroni Adjustments in Tests for Regression Coefficients. Multiple Linear Regression Viewpoints, 32(1), 1–10.).”

Thank you for including the reference. The paper does not demonstrate what the authors claim it does. Because the authors have not specified the number of predictors used to generate their Table 4, I cannot refer to a particular row from Mundfrom et al. Table 1. Tentatively, I use those with eight predictors. For a sample size of 300, the Type I error rate is not 0.05 but 0.3249! Thus, some adjustment is needed. As the authors write, however, the Bonferroni adjustment is overly conservative. With two of the eight significant (i.e., evaluating Type II error), the Bonferroni-adjusted rate is 0.0356, not 0.05. That is why I had suggested the Holm-Bonferroni method. Any stepwise method for which simulations have shown adjustment to close to the nominal rate seems appropriate. No adjustment is not suitable based on the reference that the authors provided.

Answer: We added the missing information and as strongly suggested, we have added the adjusted p-Value (Holm-Bonferoni) to table 4 and have changed the result part accordingly:

“To identify predictive variables for procedure cancellation, a binary logistic regression model which included three parameters, was used.” […]. A strong predictor of cancellation was the involvement of a second anesthesiologist during PAC, which was associated with an almost fourfold increase in cancellation risk (OR: 3.823; 95% CI: 2.035–7.182; p = 0.003). By contrast, medium cardiac risk score or if the PAC was already a second PAC was not significantly associated with an increased risk of cancellation in the analysis (OR: 1.585; 95% CI: 1.022–2.456; p = 0.078 and OR: 1.316; 95% CI: 0.644–2.689; p = 0.451).

Comment 3. I wrote “I question lines 32 and "To ensure optimal use of anesthesiologic resources, it is essential to avoid PACs for patients who will not undergo surgery or intervention." Figure 3A shows that the lowest cost solution is often no surgery.” The authors state “Of note, the aim of this study was not to determine which treatment approach leads to the lowest cost but to elute the current PAC situation and cancellation reasons.” Then, the authors’ conclusion is not based on the study. I am quoting the last sentence of the abstract.

Answer: The reviewer states " that the lowest cost solution is often no surgery". We cannot see how Fig. 3A leads to this conclusion. In the Figure the proportions of cancelation reasons are shown. Based on the Figure is it not possible to determine the cost of treatment. If the reviewer referred to the small proportion of PACs in the NoANE group compared to the total study population we understand the point. Therefore, we made the statement in line 32 more specific.

Comment 4. I gave the authors the search protocol:

 ("preoperative evaluation" OR "preanesthesia evaluation") ("not leading to procedures" OR "did not lead to subsequent procedures" OR "procedure was not performed" ). Please address the related articles. The authors explain that “The focus on the anesthesiological aspect of the evaluation is rather underrepresented.” The authors’ point is reasonable. Consider the narrower search:

("preoperative evaluation" OR "preanesthesia evaluation") ("not leading to procedures" OR "did not lead to subsequent procedures" OR "procedure was not performed" ) ("anesthesiologists" OR "anaesthesiologists" OR "anesthetists" OR "anaesthetists")

The first two articles obtained seem related, one for veterinary anaesthesia and the other for human.

Answer: We thank the reviewer for the suggestion and added both articles to the references in the revised manuscript.

Regarding the Shapiro-Wilk test, please add a sentence or phrase explaining how the results were applied. If the authors consistently apply one test or the other based on the result, then say that. Please see that this is not stated at line 106. Consider simply deleting the sentence about the Shapiro-Wilk test

Answer: We added this sentence to the static paragraph in the method section to clarify the consequences of the Shapiro Wilk test “As none of the data sets were normally distributed only non-parametric tests were used for analysis.”
